# Adhesions in abdomino-pelvic surgeries: A real economic impact?

**Perrine Capmas**[1☺], **Florent Payen**[2☺], **Anais Lemaire**[3☺], **Hervé Fernandez**[1☺]*

**1** Department of Gynecology Obstetrics, Hospital of Kremlin-Bicêtre, University Paris-Saclay, Le Kremlin Bicêtre, France, **2** Baxter SAS, Guyancourt, France, **3** Consultants in Quantitative Methods: CQM, Issy-Les-Moulineaux, France

☺ These authors contributed equally to this work.
* herve.fernandez@aphp.fr

**Data Availability Statement:** All relevant data are within the paper and its Supporting Information files.

## Abstract

### Study objective

To evaluate the epidemiologic and economic burden related to adhesions and their complications for the French healthcare system.

### Design

A descriptive and economic retrospective analysis.

### Setting

Medicalized information system program (PMSI), national scale of costs.

### Patients

Female patients operated on to treat adhesions related complications in 2019.

### Interventions

All patients with coded adhesiolysis acts were selected in order to identify the characteristics of Diagnosis related groups (DRG) and compare them with the general DRGs. Then, a sub-analysis on surgery types (laparoscopy or open procedures) was performed to evaluate impact adhesions development and Length of Stay. Lastly, direct costs of adhesions for the healthcare system were quantified based upon adhesiolysis acts coded as main diagnosis.

### Measurements and main results

26.387 adhesiolysis procedures were listed in France in 2019 through 8 adhesiolysis acts regrouping open surgeries and laparoscopic procedures. Adhesiolysis was coded in up to 34% in some DRGs for laparoscopic procedures. 1551 (1461 studied in our study) surgeries have been realized in 2019 with main procedure: adhesiolysis. These surgeries were associated with an expense of €4 million for the healthcare system for rehospitalizations and reoperations only. Social costs such as sick leaves, drugs and other cares haven't been taken in consideration.

**Funding:** The authors received no specific funding for this work.

**Competing interests:** I have read the journal's policy and the authors of this manuscript have the following competing interests: Baxter company for 2 authors This does not alter our adherence to PLOS ONE policies on sharing data and materials

## Conclusion

Adhesions related complications represent a massive burden for patients and an expensive problem for society. These difficulties may likely to be reduced by a broader use of antiadhesion barriers, at least in some targeted procedures.

## Introduction

Adhesions are one of the most common cause of complications after abdomino-pelvic surgery and cause short and long-term complications such as female infertility, chronic pelvic pain and small bowel obstruction (SBO) all lifelong [1].

Additionally, adhesions induce reoperations, hospital readmissions and consequently, a significant expenditure for the healthcare system and for society in terms of work lost, force capacity and impaired quality of life [2]. In North America and Europe, 40 to 70% of scheduled interventions for general surgery are re-operations [3]. Krielen et al. lastly found, in the SCAR update study, that laparoscopy directly reduces the risk of adhesion related rehospitalization by 32% and by 11% for possibly related rehospitalization [4].

Despite the progress in terms of variety of commercially available anti-adhesion barriers, both surgeons and healthcare administrators remain unconvinced that the current evidence for adhesion prevention products warrants routine use [5]. Moreover, even with the advances in surgical techniques in recent years, the burden of adhesions-related complications has not changed [6]. Adhesions generate major clinical, social and economic burden still.

Although all costs related to adhesions are difficult to quantify, their costs are estimated being between 60 and 600 million € per year in France [1], showing the considerable contribution that adhesions make on the resources and health expenditure.

Available economic data being rare on the cost of adhesions, we have decided to explore the PMSI (medicalized information system program) database to discuss the interest of anti-adhesion products and their use.

## Materials and methods

This retrospective descriptive study following by economic evaluation was conducted in France through PMSI database to evaluate adhesions epidemiology and their associated direct costs for hospitals and the social security (the French public health insurance). Through this analysis, we have only focused our researches on the rehospitalization rate and their induced costs. Other important costs related to adhesions consequences such as the analgesic costs, the induced infertility leading to *in-vitro* fertilization or the sick leaves cannot be found via this database. Therefore, this analysis shows only a part of costs directly related to abdomino-pelvic adhesions (synechia were excluded) and their consequences on a cohort of patients.

PMSI provides a synthetic and standardized description of the medical activity of public and private health establishments, which is based on the recording of standardized medico-administrative data in a standard collection of information. All data for this study are public and have been extracted from Scan Santé website, which is managed by Agency for Information on Hospital Care (ATIH) from the native PMSI database. ATIH provides on this site information about public and private hospital medical activities, and especially information about the Diagnosis Related Groups which is made up medical acts and diagnosis.

First, all adhesiolysis Common Classification of Medical Acts (CCAM) acts carried out in France in 2019 in public and private hospitals have been extracted. Then, for each act previously obtained, the 10 main DRG have been identified. From these DRG, information such as number and Length of Stay (LoS) were available. These different data have been put in perspective with those of general DRG. This comparison allowed to quantify the direct part of adhesiolysis in the French hospital system according to surgical procedures. We performed a sub-analysis to evaluate if the type of procedures (open surgeries vs laparoscopic approaches) had an impact on the presence of adhesions and patient's LoS. For this part, the CCAM acts falling under the same type of procedures have been grouped in order to facilitate the analysis of data, i.e that all laparoscopy acts and all open surgeries acts were grouped together.

Furthermore, we had a specific look at adhesiolysis as a main diagnosis (N736 PMSI Code). From that, we were able to obtain a DRG list providing adhesiolysis as the main reason of patient hospitalization. In order to give an idea on the direct costs of re-operation related to adhesions, we extracted the cost for each DRG from the 2018 costs national scale which is a scale providing a calculated real average cost for each DRG. In our data extraction, the origin of adhesion was not identifiable.

## Results

26.387 adhesiolysis procedures were listed in France in 2019 through 8 CCAM acts (HPPC001, HPPC002, HPPC003, HPPA001, HPPA002, HPPA003, HGPA004 and HGPC015) regrouping open surgeries and laparoscopic procedures (Table 1). This adhesiolysis number is certainly minimized as surgeons can only code two procedures per patient in the PMSI. Therefore, adhesiolysis is rarely rated especially if it is part of a combined procedure.

Regarding the HPPC001 and HPPC002 laparoscopic surgical acts grouping, adhesiolysis acts raise 34% and 27% in the 13C191 and 13C19J DRGs respectively, which are sterility procedures or grounds related to reproductive care either on an outpatient basis or level 1 (Fig 1). We found also an important number of adhesiolysis acts in the 13C07J (Intervention on the uteroannexal system for non-malignant conditions, other than tubal interruptions, on an outpatient basis) and 13C071 (Intervention on the uteroannexal system for non-malignant conditions, other than tubal interruptions, level 1) DRGs, i.e 2008 and 2775 respectively what

**Table 1. Descriptive of different CCAM adhesiolysis acts in the PMSI in 2019.**

| CCAM acts | Wording | Acts number in 2019 |
|---|---|---|
| HPPC001 | Release of neither extensive nor tight adhesions [Adhesiolysis] of the pelvic peritoneum for sterility in women, by laparoscopy | 2451 |
| HPPC002 | Release of extensive and/or tight adhesions [Adhesiolysis] of the pelvic peritoneum for sterility in women, by laparoscopy | 6158 |
| HPPC003 | Section of bridle and/or peritoneal adhesions for acute bowel obstruction, by laparoscopy | 4482 |
| HPPA001 | Release of neither extensive nor tight adhesions [Adhesiolysis] of the pelvic peritoneum for sterility in women, by laparotomy | 265 |
| HPPA002 | Section of bridle and/or peritoneal adhesions for acute bowel obstruction, by laparotomy | 5268 |
| HPPA003 | Release of extensive and/or tight adhesions [Adhesiolysis] of the pelvic peritoneum for sterility in women, by laparotomy | 755 |
| HGPA004 | Extended release of the small bowel [Extended enterolysis] for acute obstruction, by laparotomy | 4881 |
| HGPC015 | Extended release of the small bowel [Extended enterolysis] for acute obstruction, by laparoscopy | 2127 |

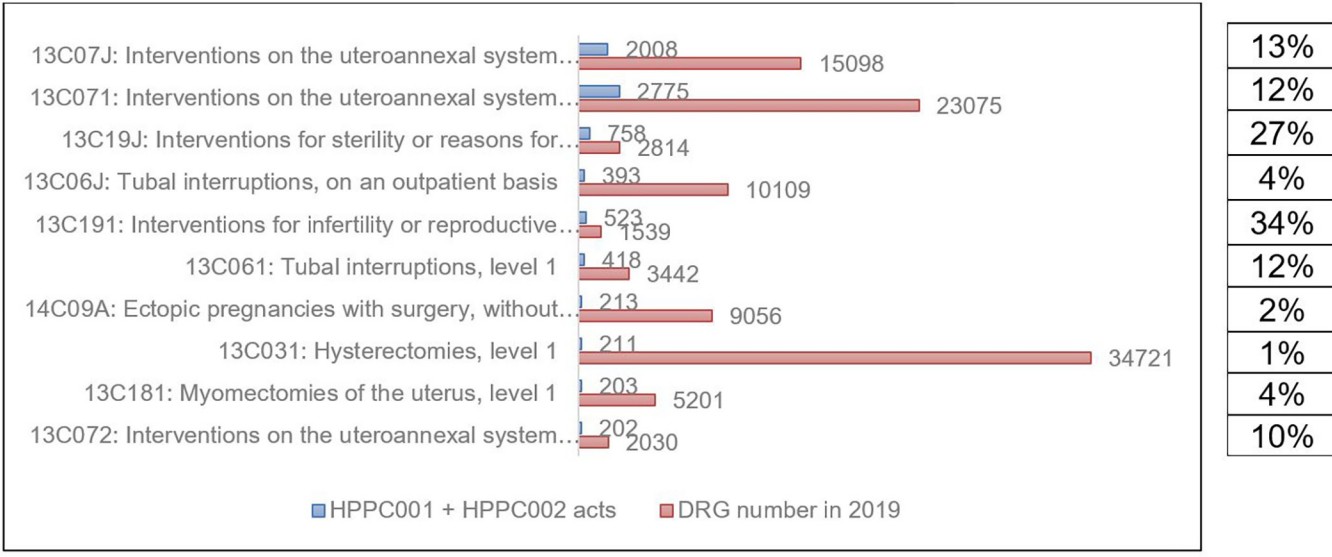

**Fig 1. Main DRG according to HPPC001 and HPPC002 medical acts and percentage of both acts in the general DRG.**

represent 13% and 12% coded acts in the general DRGs. Adhesiolysis acts by the open surgical procedures (HPPA001 and HPPA003) are less represented in the DRG, i.e found in 5% of 13C181 (Myomectomy of the uterus, level 1) and 13C182 (Myomectomy of the uterus, level 2) DRGs (Fig 2).

This analysis showed that presence of adhesions increased mean LoS by about one day for open procedures (Table 2) while no LoS change has been observed compared to average LoS of the general DRG in the laparoscopy group (Table 3) which might be related to the well-known benefits of laparoscopy.

The rest of this study deals with the N736 main and associated diagnosis coding for pelvi-peritoneal adhesions in women. N736 has been coded 1551 times as main diagnosis in 2019 and much more number as associated diagnosis.

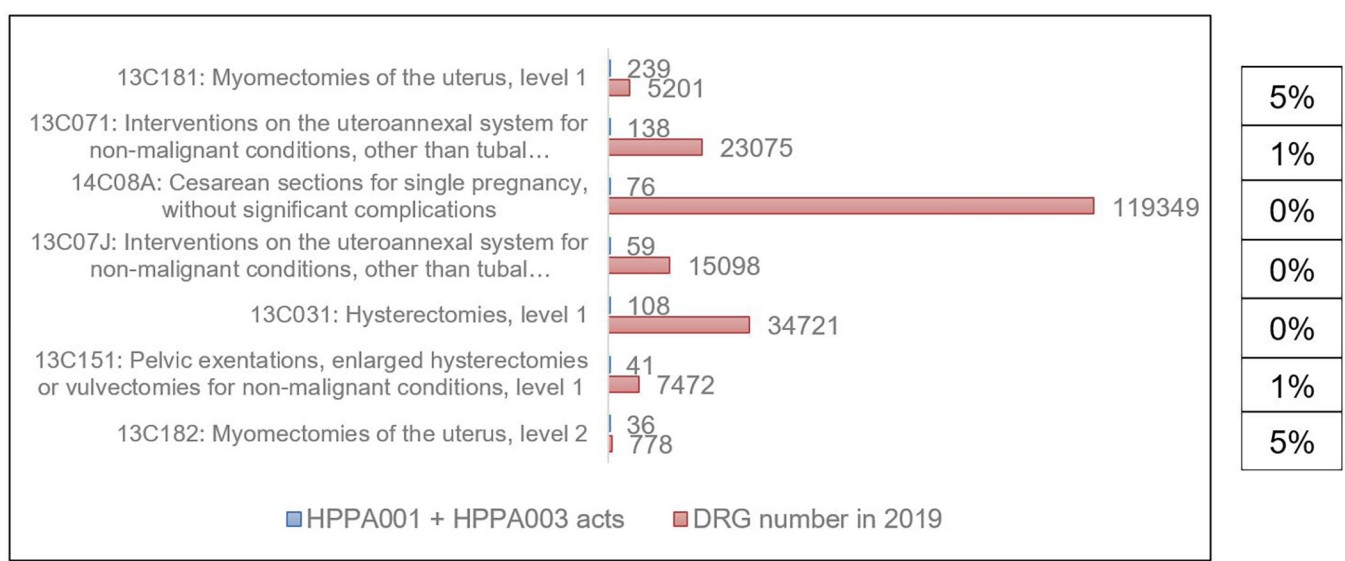

**Fig 2. Main DRG according to HPPA001 and HPPA003 medical acts and percentage of both acts in the general DRG.**

**Table 2. Main DRG according to HPPA001 and HPPA003 medical acts.**

| DRG wording | HPPA001 + HPPA003 weighted average LoS | DRG average LoS |
|---|---|---|
| 13C181: Myomectomies of the uterus, level 1 | 4,10 | 2,94 |
| 13C071: Interventions on the uteroannexal system for non-malignant conditions, other than tubal interruptions, level 1 | 2,89 | 1,89 |
| 14C08A: Cesarean sections for single pregnancy, without significant complications | 5,63 | 5,29 |
| 13C07J: Interventions on the uteroannexal system for non-malignant conditions, other than tubal interruptions, on an outpatient basis | 0 | 0 |
| 13C031: Hysterectomies, level 1 | 4,08 | 2,69 |
| 13C151: Pelvic exenterations, enlarged hysterectomies or vulvectomies for non-malignant conditions, level 1 | 3,76 | 3,12 |
| 13C182: Myomectomies of the uterus, level 2 | 6,08 | 5,07 |

Direct cost of adhesion-related re-operations reached almost €4 million even though this is a very conservative model since the analysis has been calculated on an effective of 1461 (instead of 1551 really records). Indeed, due to statistical secrecy putting place by ATIH, it is not possible to access to DRG when the number of main diagnosis is less than 11 cases (Table 4).

## Discussion

Postoperative adhesions would occur following abdominal surgery by more than 90% of patients [7]. In the case of a "second look" laparoscopy a few weeks after an abdominal procedure, incidence of adhesions is 50 to 100% [8]. Furthermore, after gynecological surgery, literature reports up to 60 to 90% of patients developing adhesions [1].

From a medical perspective, the main issue is not adhesions themselves but rather their serious consequences on the patient's post-operative course and quality of life. Almost every patient will develop adhesions after surgery and the main concern is for those who will experience complications such as female infertility, chronic pain and bowel obstruction.

**Table 3. Main DRG according to HPPC001 and HPPC002 medical acts.**

| DRG wording | HPPC001 + HPPC002 weighted average LoS | DRG average LoS |
|---|---|---|
| 13C07J: Interventions on the uteroannexal system for non-malignant conditions, other than tubal interruptions, on an outpatient basis | 0 | 0 |
| 13C071: Interventions on the uteroannexal system for non-malignant conditions, other than tubal interruptions, level 1 | 1,88 | 1,89 |
| 13C19J: Interventions for sterility or reasons for reproductive care, on an outpatient basis | 0 | 0 |
| 13C06J: Tubal interruptions, on an outpatient basis | 0 | 0 |
| 13C191: Interventions for infertility or reproductive care grounds, level 1 | 1,52 | 1,46 |
| 13C061: Tubal interruptions, level 1 | 1,88 | 1,76 |
| 14C09A: Ectopic pregnancies with surgery, without significant complications | 1,41 | 1,47 |
| 13C031: Hysterectomies, level 1 | 2,59 | 2,69 |
| 13C181: Myomectomies of the uterus, level 1 | 1,96 | 2,94 |
| 13C072: Interventions on the uteroannexal system for non-malignant conditions, other than tubal interruptions, level 2 | 5,04 | 5,09 |

**Table 4. Direct cost evaluation for reoperations related to adhesions.**

| DRG number | DRG Wording | Main diagnosis number = N736 | Estimative DRG cost 2018 (€) | Adhesions cost when DP = N736 |
|---|---|---|---|---|
| 13C07J | Interventions on the uteroannexal system for non-malignant conditions, other than tubal interruptions, on an outpatient basis | 622 | 2449 | 1 523 278 |
| 13C071 | Interventions on the uteroannexal system for non-malignant conditions, other than tubal interruptions, level 1 | 538 | 3058 | 1 645 204 |
| 13C091 | Diagnostic Laparoscopies or coelioscopies, level 1 | 70 | 2148 | 150 360 |
| 13C09J | Diagnostic Laparoscopies ou coelioscopies, on an outpatient basis | 69 | 1726 | 119 094 |
| 13C061 | Tubal interruptions, level 1 | 44 | 2978 | 131 032 |
| 13C13T | Other intervention on the female genital, very short duration | 35 | 1838 | 64 330 |
| 13C06J | Tubal interruptions, on an outpatient basis | 26 | 2483 | 64 558 |
| 13C131 | Other intervention on the female genital, level 1 | 23 | 3939 | 90 597 |
| 13C072 | Intervention on uteroannexal system for non-malignant conditions, other than tubal interruptions, level 2 | 20 | 4920 | 98 400 |
| 13C031 | Hysterectomies, level 1 | 14 | 4037 | 56 518 |
| | **TOTAL** | **1461 (1551\*)** | / | **3 943 371** |

Study concerns 1461 DRG cases as it is not possible to access data when number is less than 11 cases due to statistical secrecy.

Although adhesive SBO is relatively rare, SBO is the most severe complication of postoperative adhesions with a 10% risk of mortality [9, 10]. SBO is associated with substantial morbidity such as long hospitalization (mean length of stay: 8 days) and in-hospital mortality rate of 3% per episode [11–14].

Regarding chronic abdominal pain and adhesions, the relationship between both is complex and subject to debate [15]. Chronic pelvic pain is a problem for 20 to 50% of women with adhesions and may lead to psychological disorders, depression, and even suicide [16].

Adhesions are the first cause of secondary female infertility and are found in approximately 20–40% during laparoscopic evaluation for acquired female infertility [17].

Adhesions can affect the quality of life of millions of patients, jeopardize life expectancy, and result in more than US$2 billion dollars of healthcare costs in the USA yearly [3].

Through this study, we have attempted to evaluate the cost directly related to adhesions (mainly complications, reoperations and rehospitalization) for the French healthcare system. This easy calculable cost raises in 2018 to €4 million and seems to be a tiny fraction of real societal costs related to adhesions, i.e the visible face of adhesions cost expenditures. Beyond that, it is important to mention that the national costs scale can provide some biases on the real cost of DRGs and remain also an estimative cost. Establishments taken in account in the annual development of this cost national scale are volunteers and represent only a sampling of all French establishments which could lead to costs estimation errors either higher or lower.

In our study, we were not able to identify if the adhesions were related to a previous surgery or an infection. Regardless their origin, adhesions will have the same clinical impact, therefore prevention of adhesions related complications remains a key subject. In order to reduce adhesion formation and prevent adhesion related complications, in addition to surgical techniques, various products with different mechanisms of action exist on the market. These products can be solid membranes, gels and liquids. Although the side effects seem rare in the use of these products, the lack of evidence has often posed major obstacles in non-use of these product despite of large number of positive evidences published on the subject. To date, there is no published randomized controlled trial comparing these products. Efficacy of antiadhesion barriers in open surgery has been well established for reducing the incidence of adhesion

formation during the past years and their routine use would be deemed cost-effective. Regarding the laparoscopic surgery, antiadhesion barriers may be cost-effective, especially in fertile-age female patients where the direct costs of complications related to adhesions for the health-care system reached 1320$ (970$ in all patients) [17]. Benefits of anti-adhesion barriers after laparoscopic myomectomy were recently proven by a literature review and a metanalysis in which authors concluded that use of cellulose absorbable barrier reduced the risk of postoperative adhesions and that some adhesions barriers methods seemed to show results in reducing the incidence of adhesions [18, 19]. De Wilde et al. indicated in 2007 that surgeons should actively consider adopting anti-adhesion strategies, particularly in "high risk" gynecological procedures such as ovarian, endometriosis or tubal surgery, myomectomy and adhesiolysis [6]. Furthermore, following our results, it seems legitimate to think that an important part of expenditures could be avoidable with a broader use of the antiadhesion barriers. Those products might improve patient's post-operative course, quality of life, and decrease the societal cost related to adhesion. Besides these questions and despite the increasing published articles showing their efficacy in various procedures such as myomectomy, ovarian cystectomy or surgeries to treat endometriosis, antiadhesion barriers remain often considered as too expensive and not perceived as a cost-effective tool. And yet, these surgeries, due to their conservative nature are responsible for pain, infertility and bowel obstruction leading to identified over costs compared to drastic surgeries for which the infertility cost, for example, doesn't appear. That sounds reasonable to democratize the access and use to the anti-adhesion barriers [3, 14, 17] in these specific surgeries at least and much more in female fertile-age patients [17].

## Conclusion

Our results demonstrate that the extra-cost directly related to post-operative adhesions is massive–even though this is a very conservative model since this analysis doesn't include the additional non-surgical costs (social costs, drugs. . .). Various products available on the market have demonstrated their efficacy in reducing these adhesions and their complications. Therefore, it seems reasonable to make every effort in reducing post-operative adhesions, both for medical and medico-economic considerations.

Our study is a local and retrospective analysis and under-estimates costs. It would be valuable to perform a larger medico-economic study to evaluate more precisely the accurate rapport between cost of adhesions complications and cost of adhesion reduction agents. But the significant cost of adhesions-related complications demonstrated in our analysis supports the cost-effectiveness properties of adhesions barriers, and advocates for a broader use of these products.

## Supporting information

**S1 Data.**
(XLSX)

## Author Contributions

**Conceptualization:** Perrine Capmas, Florent Payen, Anais Lemaire, Hervé Fernandez.

**Methodology:** Perrine Capmas, Florent Payen, Anais Lemaire, Hervé Fernandez.

**Validation:** Perrine Capmas, Florent Payen, Anais Lemaire, Hervé Fernandez.

**Writing – original draft:** Perrine Capmas, Florent Payen, Anais Lemaire, Hervé Fernandez.

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
