## [Decision Letter · Decision Letter 0]

8 Mar 2022

PONE-D-21-36502Adhesions in abdomino-pelvic surgeries: a real economic impact?PLOS ONE

Dear Dr. Fernandez,

Thank you for submitting your manuscript to PLOS ONE. After careful consideration, we feel that it has merit but does not fully meet PLOS ONE’s publication criteria as it currently stands. Therefore, we invite you to submit a revised version of the manuscript that addresses the points raised during the review process.

Please revise the manuscript according to reviewers's suggestions.

We look forward to receiving your revised manuscript.

Kind regards,

Diego Raimondo

Academic Editor

PLOS ONE

Journal Requirements:

"I have read the journal's policy and the authors of this manuscript have the following competing interests: Baxter company for 2 authors"

We note that you received funding from a commercial source: Baxter Company

Reviewers' comments:

Reviewer's Responses to Questions

**Comments to the Author**

1. Is the manuscript technically sound, and do the data support the conclusions?

Reviewer #1: Yes

Reviewer #2: No

Reviewer #3: Yes

2. Has the statistical analysis been performed appropriately and rigorously? 

Reviewer #1: Yes

Reviewer #2: I Don't Know

Reviewer #3: Yes

3. Have the authors made all data underlying the findings in their manuscript fully available?

Reviewer #1: Yes

Reviewer #2: Yes

Reviewer #3: Yes

4. Is the manuscript presented in an intelligible fashion and written in standard English?

Reviewer #1: Yes

Reviewer #2: Yes

Reviewer #3: Yes

5. Review Comments to the Author

Reviewer #1: Dear Authors,

thank you for submitting your manuscript to “Plos One” journal.

I think the topic of is of outstanding interests for both surgeons and healthcare system, in particular in developed countries. The manuscript offers some interesting insights in the health and economic costs of adhesions management.

Nevertheless, I have the follow comments:

- Why do you think length of hospital stay (LoS) in significantly higher for open surgeries with adhesiolysis, and not for laparoscopic procedures including adhesiolysis as well? Do you hypothesize possible reason for this discrepancy, or is this due to the well-known benefits of laparoscopy?

- As the authors admit in the “conclusion” section of the manuscript, costs of adhesions are probably underestimated by this study, that considers only direct surgical costs. Further studies are advocated to include also surgical procedures complicated by adhesiolysis, even if it is not coded as the main surgery, and indirect costs (drugs, work absenteeism, ecc)

- Lines 201-203: benefits of anti-adhesion barriers after laparoscopic myomectomy were recently proven by a literature review and a metanalysis: see

Adhesion barriers in laparoscopic myomectomy: Evidence from randomized clinical trials. Borghese G, et al. Int J Gynaecol Obstet. 2021. PMID: 33237574

Eur J Obstet Gynecol Reprod Biol 2020; doi: 10.1016/j.ejogrb.2019.12.033. Cellulose absorbable barrier for prevention of de-novo adhesion formation at the time of laparoscopic myomectomy: A systematic review and meta-analysis of randomized controlled trials. Raimondo D, et al. Eur J Obstet Gynecol Reprod Biol 2020; doi: 10.1016/j.ejogrb.2019.12.033.

- Some references should be updated

- Some minor English mistakes should be corrected.

Reviewer #2: This is a nice and well written manuscript

However, I think it add few data to what is already known.

I also suggest to review the manuscript in terms of language and grammar.

Please also provide high resolution figures

Reviewer #3: Interesting manuscript and great analysis, The authors make a great point regarding adhesion prevention during surgery. This paper could inspire other authors to conduct research on this very important topic. I believe that this manuscript is ready fro publication.

6. PLOS authors have the option to publish the peer review history of their article (what does this mean?). If published, this will include your full peer review and any attached files.

Reviewer #1: No

Reviewer #2: No

Reviewer #3: No

---

## [Author Response · Author response to Decision Letter 0]

27 Sep 2022

Dear Reviewers,

We would like to thank you regarding your interest, reading time and confidence about our manuscript entitled “Adhesions in abdomino-pelvic surgeries: a real economic impact?”. We are delighted to read our work raised your attention and comments in order to increase quality of manuscript. Hereafter, you can find responses to your raised comments:

- Rewiever #1:

o Regarding Length of hospital stay which is higher for open surgeries compared to laparoscopic procedures, we hypothesize differences are due to the well-known benefits of laparoscopy. 

o We confirm that costs of adhesions are probably underestimated since indirect costs are not included. Further studies would be interesting to produce.

o References about benefits of anti-adhesion barriers after laparoscopic myomectomy have been well added in our manuscript which confirms interest of anti-adhesion barriers in all surgical approaches.

- Reviewer #2: No comment due to absence of question.

- Reviewer #3: No comment due to absence of question.

Furthermore, no funding was received from Baxter for any author of this paper. Please not that Dr Payen and Dr Lemaire were Baxter employees when the article was written, but they received no additional funding for this work apart of their monthly salary. Pr Fernandez and Dr Capmas did not receive any financial support from Baxter for this work.

Hoping that revised version will allow a future publication in “Plos one” review.

Yours faithfully.

---

## [Editor Report · Decision Letter 1]

14 Oct 2022

Adhesions in abdomino-pelvic surgeries: a real economic impact?

PONE-D-21-36502R1

Dear Dr. Fernandez,

We’re pleased to inform you that your manuscript has been judged scientifically suitable for publication and will be formally accepted for publication once it meets all outstanding technical requirements.

Kind regards,

Diego Raimondo

Academic Editor

PLOS ONE

---

## [Editor Report · Acceptance letter]

18 Oct 2022

PONE-D-21-36502R1 

Adhesions in abdomino-pelvic surgeries: a real economic impact? 

Dear Dr. Fernandez:

I'm pleased to inform you that your manuscript has been deemed suitable for publication in PLOS ONE. Congratulations! Your manuscript is now with our production department. 

Kind regards, 

on behalf of

Dr. Diego Raimondo 

Academic Editor

PLOS ONE